# Human NQO1 as a Selective Target for Anticancer Therapeutics and Tumor Imaging

**DOI:** 10.3390/cells13151272

**Published:** 2024-07-29

**Authors:** A. E. M. Adnan Khan, Viswanath Arutla, Kalkunte S. Srivenugopal

**Affiliations:** Department of Pharmaceutical Sciences, Jerry H. Hodge School of Pharmacy, Texas Tech University Health Sciences Center, 1406 Amarillo Research Bldg., Rm. 1102, Amarillo, TX 79106, USA; aekhan@ttuhsc.edu (A.E.M.A.K.); viswanatharutla@gmail.com (V.A.)

**Keywords:** antioxidant enzymes, tumor-selective therapies, β-lapachone, futile substrates, theranostic drugs, targeted therapy, tumor imaging, NQO1 turn-on probes, near-infrared fluorophores

## Abstract

Human NAD(P)H-quinone oxidoreductase1 (HNQO1) is a two-electron reductase antioxidant enzyme whose expression is driven by the NRF2 transcription factor highly active in the prooxidant milieu found in human malignancies. The resulting abundance of NQO1 expression (up to 200-fold) in cancers and a barely detectable expression in body tissues makes it a selective marker of neoplasms. NQO1 can catalyze the repeated futile redox cycling of certain natural and synthetic quinones to their hydroxyquinones, consuming NADPH and generating rapid bursts of cytotoxic reactive oxygen species (ROS) and H_2_O_2_. A greater level of this quinone bioactivation due to elevated NQO1 content has been recognized as a tumor-specific therapeutic strategy, which, however, has not been clinically exploited. We review here the natural and new quinones activated by NQO1, the catalytic inhibitors, and the ensuing cell death mechanisms. Further, the cancer-selective expression of NQO1 has opened excellent opportunities for distinguishing cancer cells/tissues from their normal counterparts. Given this diagnostic, prognostic, and therapeutic importance, we and others have engineered a large number of specific NQO1 turn-on small molecule probes that remain latent but release intense fluorescence groups at near-infrared and other wavelengths, following enzymatic cleavage in cancer cells and tumor masses. This sensitive visualization/quantitation and powerful imaging technology based on NQO1 expression offers promise for guided cancer surgery, and the reagents suggest a theranostic potential for NQO1-targeted chemotherapy.

## 1. Introduction

NAD(P)H:quinone oxidoreductases (NQOs) comprise a group of flavoproteins catalyzing the two-electron reduction of quinones and their derivatives. In mammalian systems, there exist two NQO members: NQO1 and NQO2. NQO1 utilizes NAD(P)H, while NQO2 employs dihydro nicotinamide riboside (NRH) as the electron donor. Beyond their well-documented role in reducing quinone compounds and inhibiting reactive oxygen species formation, NQOs, particularly NQO1, have multifaceted functions in eliciting anti-inflammatory effects, the direct scavenging of superoxide anion radicals, the stabilization of p53 tumor suppressor protein, cancer therapy, and tumor imaging [1]. NQO1 was first found in rodent livers and was given the name DT-diaphorase [2]. In contrast, the identification and characterization of NQO2 involved screening a human liver cDNA library and hybridization with an NQO1 cDNA probe [3,4]. Mammalian tissues exhibit little to moderate expression of both NQO1 and NQO2, with the liver, kidney, and cardiovascular systems displaying moderate levels [2,4,5]. In humans, *NQO1* and *NQO2* are located on chromosomes 16q22.1 and 6pter-q12, respectively [2,6]; these proteins are cytosolic, with molecular weights of 31- and 25-kDa, respectively [2]. NQO1 and NQO2 share considerable similarities in protein sequence, structure, and catalytic mechanism, except that they reduce different quinone substrates using different cofactors (NADPH vs. N-ribosyldihydronicotinamide). The inhibitors for each enzyme are also distinct. Their crystal structures have allowed a structural and functional comparison, facilitating a deeper understanding of the two enzymes [4,7,8,9,10,11,12]. Therefore, it is expected that NQO1 and NQO2 cooperate in cellular quinone detoxification and may compensate for each other. In the context of this review dealing with the bioactivation of therapeutic quinones and tumor imaging, this is an important consideration. However, only NQO1 has been widely investigated and will be highlighted throughout the narration.

While numerous genes participate in antioxidant defense, only a few selected ones possess the capacity to influence the human response to environmental factors that induce oxidative stress. Human genes encoding enzymes, including the *GSTs* (glutathione S transferases), SOD (superoxide dismutase), *NQO1*, *CAT* (catalase), NOS (nitric oxide synthase), and *HMOX1* (heme oxygenase 1), which are regulated by the Nrf2 transcription factor in the prooxidant milieu, are such examples. NQO1 is now recognized as a versatile cytoprotector, functioning as a multifunctional antioxidant [13]. On the flip side, it has been implicated in the generation of DNA damage, the redox regulation of chromatin binding proteins, and carcinogenesis as well [14,15].

The unique physiology of tumors, namely, the elevated inherent oxidative stress that occurs due to increased metabolic demand, incomplete electron transport chain activity, the autocrine production of cytokines, and other mechanisms [16], also triggers an enhanced expression of NQO1 and other antioxidant enzymes. Targeting this stress phenotype, exclusive to human malignancies but not the normal tissue counterparts, provides unique opportunities for tumor-selective therapy and cancer imaging, as discussed here.

## 2. NQO1 Gene, Expression, Cellular Functions, and Anticancer Bioreductive Catalysis

In the early 1960s, Martius et al. explored a protein identified as vitamin K reductase, and later, Ernster et al. identified DT-diaphorase [17,18], which advanced the NQO1 enzymology. NQO1 is known to convert vitamin K to vitamin K hydroquinone but has a negligible effect on the blood clotting cycle [17]. However, vitamin K oxidoreductase (VKOR; EC1.17.4.4) catalyzes most of this transition [18]. NQO1 plays a significant role in reducing the cellular burden of free radicals by avoiding the reduction of one electron or semiquinones [19,20], and this antioxidant action is consistent with its frequent upregulation in response to cellular stress, electrophilic compounds, and xenobiotic detoxification [20,21,22,23,24,25].

The location, structure, and antioxidant regulation of the human *NQO1* gene by NRF2 and the resulting gene product are described in Figure 1. The catalytically active enzyme is a homodimer. Consistent with an antioxidant role and induction by a large number of electrophiles, carcinogens, and natural compounds [26], the gene promoters of both *NQO1* and *NQO2* contain six antioxidant response elements (AREs), which have been well characterized and their binding with the NRF2 protein established [27,28]. Both the constitutive and oxidant-induced expressions of *NQO1* are governed by the NRF2 transcription factor [19], which explains the lower levels of protein expression in unstressed normal tissues and the elevated content in their malignant counterparts. Consistent with an antioxidant role, a susceptibility to cancer development and cardiovascular and other diseases has been linked to decreased NQO1 activity [29]. Compared with the wild-type *NQO1*1* allele coding for normal levels of enzyme activity, the *NQO1*2* allele encoding a polymorphic variant at position 187 (P187S, proline to serine) with negligible catalytic activity is expressed in over 25% of the world’s population, according to the Ensembl database; the *NQO1*2* allelic frequency ranges between 0.22 (Caucasian) and 0.45 (Asian), predominating in half of the Chinese population [30]. A large epidemiologic investigation of a benzene-exposed population has shown that *NQO1*2* homozygotes exhibit as much as a 7-fold greater risk of bone marrow toxicity, leading to diseases such as aplastic anemia and leukemia [29].

Besides the generally heightened expression across the human cancer types, hypoxia, another hallmark of malignancies, has an impact on NQO1 content and function. The evidence indicates that NQO1 functions to stabilize HIF-1α (hypoxia-inducible factor-1) by preventing its degradation [31,32]. HIF-1α target genes such as *GLUT1* and *VEGF* are significantly elevated in high NQO1 expressing, indicating that HIF-1α transcriptional signaling may be activated in NQO1-positive cancers and confer a poor prognosis [31,32]. Additionally, hypoxia per se may increase the expression of NQO1 [32]. A higher NQO1 content in human malignancies has been linked to a higher burden of free radicals and redox imbalance (described later); however, the killing of pancreatic cancer cells after the inhibition of NQO1 by dicoumarol and other compounds [33,34,35,36,37,38] suggests the operation of off-target effects through multiple mechanisms.

Figure 2 represents the multiple biochemical functions and overall significance of NQO1 in protection against carcinogenesis. There are four key functions associated with this process: (i) the removal of superoxide anion radicals by acting as a scavenger for SOD, (ii) the elimination of quinone substrates by a two-electron reduction mechanism, (iii) the restoration of endogenous antioxidants like ubiquinone and α-tocopherol, and (iv) the stabilization of the suppressor proteins p53/p73.

### 2.1. Scavenging of Superoxide Radicals

NQO1 plays a direct role in antioxidant protection [33,39,40] by inhibiting the formation of reactive oxygen species (ROS) through the redox cycling of various quinones to their hydroquinones, instead of the more reactive semiquinone forms. It prevents quinones from entering the single-electron reduction catalyzed by one-electron transfer and cytochrome P450 reductases that generate semiquinone free radicals and reactive oxygen species, thereby protecting cells from oxidative damage. Other studies suggest that NQO1 can directly eliminate superoxides in a pyridine–nucleotide-dependentreaction [33,41,42]; this property may assume significance in tissues with moderate NQO1 expression, such as the blood vessels and myocardium [35,36,37]. Further, the induction of NQO1 up to 30-fold is possible in response to oxidative stress generated by UV radiation and X-rays in human cells [38,43]. Evidence for the catalytic activity of NQO1 as a superoxide reductase has been gathered by electron paramagnetic resonance spectroscopy and many superoxide-generating systems [33,36,42].

### 2.2. Reduction and Activation of Quinone Compounds and Futile Redox Cycling

The two-electron reduction of endogenous quinones and exogenous quinones (menadione), including some therapeutic agents, has been well studied. It is generally accepted that most hydroquinone derivatives can be easily conjugated with glutathione or glucuronic acid and excreted. Nevertheless, some of the hydroquinones formed by NQO1 are unstable and can autoxidize to trigger cell death through ROS-dependent mechanisms (Figure 2 and Figure 3). Three major mechanisms of cell death by hydroquinones have been proposed. This can be through (a) the direct alkylation of DNA through the activation of drugs like mitomycin C and RH1 into alkylating groups, (b) the inactivation of heat shock proteins by geldanamycin hydroquinones generated by the enzyme, and (c) in contrast to the known metabolic process of NQO1, certain naturally occurring or synthetic quinone compounds produce labile forms of hydroquinone [39,40,41,42]. As such, redox cycling β-lapachone, strptonigrin, deoxyniboquinone, and others deplete cellular NADPH and generate respective unstable hydroquinones that react with molecular oxygen, resulting in the production of two moles of superoxides, wherein the original quinones are regenerated, as shown in the middle panel of Figure 2 [43]. Because the futile substrates produce copious amounts of cytotoxic ROS only in NQO1-proficient cells, they have excellent potential as antitumor agents [33,44,45,46,47,48,49,50,51,52,53]. In this context, the impact of superoxide dismutase on the autoxidation rates of hydroquinones generated by NQO1 is noteworthy. The effect of superoxide dismutase on autoxidation might either expedite or impede the process, contingent upon the redox chemistry and stability of the hydroquinones [39,54].

### 2.3. Preservation of Endogenous Antioxidants like Ubiquinone and α-Tocopherol

NQO1 serves as a defense mechanism against undesired oxidation by preserving the reduced form of endogenous quinones, thereby safeguarding cellular membranes against lipid peroxidation [55]. Notably, both ubiquinone (CoQ) and α-tocopherol quinone (TQ), crucial lipid-soluble antioxidants, serve as substrates for NQO1 in vitro [55,56]. Generally, the reduction rate of CoQ derivatives is dependent on chain length, with molecules containing longer chains, such as CoQ9 or CoQ10, being less efficiently reduced.

### 2.4. Stabilization of Tumor Suppressor Proteins p53 and p73

An important property of NQO1 is its ability to interact with other proteins, and several of these interactions have been demonstrated to be integral to cellular function [34]. NQO1 plays crucial roles in stabilizing the p53 protein through protein–protein interactions, inhibiting the ubiquitin-independent degradation of p53 by the 20S proteasome, and increasing its half-life [56,57,58]. This mechanism may account for the lower basal levels of the p53 protein observed in the animal model of *NQO1*-null mice [48]. The p53-NQO1 association is counteracted by dicoumarol [58]. Interestingly, the less stable forms of NQO1, such as the p.P187S polymorphic variant, exhibit a decreased efficacy in stabilizing p53 [59]. The stabilization of p53 by NQO1 may confer protective effects against carcinogenesis [34,56]. Additionally, given p53’s association with the modulation of apoptosis in vascular cells, the stabilization of the p53 protein by highly expressed NQO1 in the heart and blood vessels may contribute to protection against the dysregulation of apoptosis related to cardiovascular pathophysiology [39,56]. NQO1 has been identified as a stabilizer of other proteins, including ornithine decarboxylase, p63, p73, and peroxisome proliferator-activated receptor-γ coactivator-1α (PGC1α), in an NAD(P)H-dependent manner [34,60,61,62,63]. NQO1 also engages with and regulates proteasomal components, suggesting a role in controlling protein degradation within the cell [64].

### 2.5. Cytotoxic Mechanisms of NQO1 Bioactivating Compounds

As stated in previous sections, NQO1 acts as a double-edged sword by playing crucial roles in cellular defense against oxidative stress induction by preventing the generation of ROS and at the same time catalyzing futile reductive cycles with certain quinones, leading to heightened redox stress in cancer cells. This continuous futile catalysis can deplete cellular NAD(P)H pools, alter the macromolecular metabolism, and perturb the redox balance. Such a redox imbalance can have profound implications for cell fate, influencing different cell death mechanisms. The consequences of a redox imbalance induced by NQO1 futile cycling underscore the delicate equilibrium that cells must maintain to ensure their proper function and survival. The futile cycling of quinones triggers enhanced oxidative stress through the production of ROS and hydrogen peroxide, the consequent induction of DNA breaks and the hyperactivation of poly ADP-ribosylation, mitochondrial damage, autophagy, and the activation of signaling pathways emanating from proteostasis due to endoplasmic stress, culminating in apoptotic or necrotic cell death (44, 46, and our unpublished data) as represented in Figure 3. In addition, our laboratory has obtained evidence for immunogenic cell death (ICD) after cell tumor cell exposure to NQO1 futile substrates. Briefly, the ICD entails the release of DAMPS (damage-associated molecular patterns) from cells damaged by the NQO1-activated drugs. Examples of DAMPs include calreticulin, HMGB1, ATP, ANXA1, and type I interferon. The DAMPs bind receptors and ligands on dendritic cells, which are then engulfed as antigenic determinants. Consequently, via antigen presentation, dendritic cells stimulate specific T-cell responses that kill more cancer cells. The induction of ICD eventually results in long-lasting protective antitumor immunity [65].

### 2.6. Differential Expression of NQO1 in Normal vs. Malignant Cells

Figure 4A depicts the relative expression of *NQO1* mRNA as computed by us based on FANTOM Tag expression reported in different human organs [63]. From this data, the gallbladder and kidney appear to contain moderate levels of *NQO1*, with little in the liver among normal tissues. Figure 4B, in contrast, shows the tissue data for RNA expression obtained through a Cap Analysis of Gene Expression (CAGE) generated by the FANTOM5 project in human cancers [66]. The values represent scaled tags per million (TPM), which allows for the comparison of gene expression levels across different cancers. The results provide evidence for NQO1 abundance in malignant tissues, with an average of low to high levels of expression in several cancer types. It should be noted, however, that some cancers may fail to express this antioxidant enzyme and may not be receptive to bioactivated futile drugs. Nevertheless, on average, a majority of neoplasms exhibit strong cytoplasmic immune reactivity for the NQO1 protein, with lung cancer having the highest level of expression, followed by colorectal, endometrial, stomach, and pancreatic cancers. Besides the varied levels of altered redox pathophysiology, the tumor NQO1 content may be influenced by genetic aberrations, environmental toxicity, and the tumorigenic pathways endured [67,68]. The data presented in Figure 4A,B do come with certain limitations but shed light on the putative organ toxicities associated with NQO1-activated anticancer drugs. First, given the presence of the *NQO1*2* polymorphic form with a none to reduced activity in a quarter of the world population, the mRNA content for *NQO1* is unlikely to be an accurate reflection of the active enzyme. Secondly, the moderate NQO1 levels in the kidney and gall bladder may confer at least some toxicity by the futile cycling substrates of the enzyme. Irrespective of these considerations, the overall higher expression justifies NQO1 as a unique and selective target for tumor imaging and therapy, as discussed in the sections to follow.

In contrast to many molecularly targeted therapies, β-lap showed a broad spectrum of anticancer activity [69], spurring a high level of interest among oncologists. Compared to normal tissue, NQO1 overexpression occurs up to 200-fold in >80% of non-small cell lung cancer (NSCLC), up to 100-fold in >80% of pancreatic cancer, up to 10-fold in 60% of prostate cancer, up to 10-fold in 60% of breast cancer, and up to 10-fold in 50% of colorectal cancer [70,71,72,73]. In preclinical models, β-lap-induced cancer cell death occurred across tumors in proportion to tumor NQO1 levels [74,75,76]. In normal tissues, the induction of *NQO1*, a phase 2 detoxifying enzyme, occurs rarely (e.g., with substantial exposure to polyaromatic hydrocarbons) and is short-lived (<2 h). Importantly, factors associated with ROS detoxification (and presumptively, β-lap resistance), such as catalase expression, are deficient in cancer cells relative to normal tissues, further enhancing the potential therapeutic margin of this strategy. β-Lapachone also demonstrated synergy with ionizing radiation, cytotoxic chemotherapeutics, and PARP inhibitors [77,78], rendering it potentially useful as an anticancer drug. Nevertheless, as discussed in a later section, the insolubility and toxicity of this compound have hindered its clinical application.

### 2.7. NQO1 Bioreductive Quinone Substrates Eliciting Cytotoxicity

Over the years, there has been progress in discovering and designing futile catalysis substrates for NQO1 for anticancer effects. A significant statistical correlation between NQO1 enzymatic activity and the cytotoxicity of many antitumor quinones has been established. The chemical structures of the drugs and prodrugs belonging to this category and their properties are shown in Table 1. Not every compound known in the literature has been listed. Of these, the natural compounds β-lapachone [44,75] and deoxynyboquinone [46] have been well studied and described in sufficient detail in the subsections of this review. GNQ-9 is a synthetic spiroisoindolinone quinone synthesized and characterized in our laboratory [79]. GNQ-9 was shown to be an excellent futile substrate for NQO1, with antiproliferative activity against a variety of cancer cell lines, particularly NQO1-overexpressing tumor cells, and minimal cytotoxicity towards a panel of normal cell counterparts. GNQ-9 generated ROS production via NQO1 catalysis and inhibited topoisomerase II. Interestingly, GNQ-9 passed through the blood–brain barrier and effectively eliminated the orthotopic glioma xenografts developed in nude mice [79].

The 3,7-diaminophenothiazinium-based redox cycler (PRC) and its derivatives were shown to be very good substrates for NQO1, generating a huge ROS production in cells [85]. The ensuing cell death involved phosphatidylserine externalization, loss of mitochondrial transmembrane potential, cytochrome c release, and caspase-3 activation. Coincubation with catalase achieved cell protection, whereas reductive antioxidants enhanced PRC’s cytotoxicity. The role of NQO1 in PRC bioactivation and cytotoxicity was confirmed in stably transfected MCF-7-DT15 cells [85].

The structure-based development of NQO1-directed antitumor quinones resulted in the development of RH1 [2,5-diaziridinyl-3-(hydroxymethyl)-6-methyl-1,4-benzoquinone], a methyl-substituted diaziridinyl quinone [52]. NQO1 catalysis released an aziridine alkylating species that can crosslink the DNA [97]. Both NQO1-dependent and -independent cytotoxicity in tumor cell lines have been observed for RH1 [98]. Therefore, its potential for clinical use remains questionable. EO9 (5-aziridinyl-3-hydroxymethyl-2-(3-hydroxyprop-1-enyl)-1-methylindole-4,7-dione, Apaziquone) was developed as a derivative of mitomycin C, targeting activation by NQO1 [93,94]. Because of an excellent cytotoxic profile in cell types rich in NQO1 under normoxic conditions, EO9 underwent clinical trials. However, it failed to show activity in phase II clinical trials when administered intravenously. Poor drug delivery to tumors resulting from a combination of rapid elimination and poor penetration through avascular tissues were the major factors underlying the feeble efficacy of EO9 [95]. Therefore, an intravesical administration directly into the bladder in patients with superficial transitional cell carcinoma was attempted. However, this drug has failed to receive FDA approval in the USA [66].

In a rational prodrug approach, cytotoxic anticancer drugs such as combretastatin A, podophyllotoxin, and 5-fluorouracil (5-FU) containing the parent drug, a self-destructing linker, and an NQO1-responsive trigger group were synthesized [87,88,96]. In this strategy aimed at selective delivery to NQO1-overexpressing cells, the trigger group, as a substrate for NQO1, is connected to the parent drug via a self-immolating linker. The linker promotes enzymatic cleavage of the trigger group, to release the anticancer drug in situ. This prodrug delivery plan showed antitumor activity and safety profiles that were similar to the parent drugs alone [99]. Despite this result, NQO1-targeted delivery of anticancer drugs remains a viable strategy and merits further development.

Mitomycin C (MMC), an antitumor antibiotic, is a prototype bioreductive drug employed to treat a variety of cancers. It requires a reductive activation to be converted to a bis-electrophile that forms several covalent adducts with DNA, including an interstrand crosslink which is considered to be the lesion responsible for cytotoxicity [67,68,91]. Although good evidence exists for the activation of MMC by NQO1, there is controversy about the importance and magnitude of this effect [92]. Streptonigrin (STN36) is a moderate NQO1 substrate [89,100]. Nevertheless, due to various factors, such as activation by other reductases for MMC [92,101,102], RH1 [98,103], and STN, detoxification by NQO1 under specific conditions for EO9 [93,94], short in vivo half-lives for RH1 [97], EO9 [104], and β-lap [105], or severe toxicity (STN36), none of these compounds has been able to demonstrate the full potential of NQO1-activated anticancer agents.

### 2.8. Inhibitors of NQO1 Catalytic Activity

The concept of inhibiting NQO1 activity for cancer therapy has not been attractive because of its undefined role in cell proliferation; however, it has garnered some attention. NQO1 inhibitors are mostly used in experimental settings, and Figure 5 shows the structures known to curtail the activity of the enzyme. The chemical classes of natural NQO1 inhibitors are coumarins, flavonoids, and triterpenoids. Dicoumarol, an oral anticoagulant, is a natural hydroxycoumarin. Dicoumarol is the most commonly used inhibitor of NQO1, and it acts through competitive binding with NAD(P)H and thereby prevents the two-electron transfer to FAD from occurring [106]. High protein binding and off-target effects compromise the efficacy of dicoumarol in curtailing NQO1 activity. Among the other inhibitors listed, scopoletin and umbelliferone induce the strongest inhibitory effects compared to coumarin and aesculetin. All these compounds appear to act through a non-competitive inhibition model [107]. Diminutol is a cell-permeable 2,6,9-trisubstituted purine analog that competitively inhibits the enzyme [108]. In contrast, ES-936 (5-methoxy-1,2-dimethyl-3(4-nitrophenoxymethyl) indole-4,7-dione) is a highly specific potent, mechanism-based NQO1 inhibitor [109]. ES936 undergoes two-electron reduction by NQO1, giving the corresponding hydroquinone and the indole nitrogen lone pair, which is no longer conjugated with the quinone carbonyl group, and, hence, “normal” indole reactivity takes over, resulting in the elimination of the nitro-aryloxy group to generate a highly electrophilic iminium ion. Because this occurs in the enzyme active site, the generation of iminium leads to irreversible alkylation of the enzyme, shown by mass spectrometric studies to be at Tyr-127 or Tyr-129, thereby inhibiting the enzyme. On the cancer front, a dual inhibitor of NQO1 and glutathione transferases, called MNPC (5-methyl-N-(5-nitro-thiazol-2-yl)-3-phenylisoxazole-4-carboxamide), has been synthesized and characterized [110]. A preferential efficacy of MNPC against EGFRvIII mutant glioblastoma was observed.

### 2.9. β-Lapachone and Deoxynyboquinone, Two Potent Futile Cycling Substrates of NQO1

Over the years, β-Lapachone, an ortho-naphthoquinone natural product isolated from the lapacho tree (Tabebuia avellanedae), has received much attention for its medicinal properties and pharmacological activities, including the inhibition of topoisomerase 1 [111]. This compound has been investigated extensively as an NQO1-mediated redox cycle substrate. The sequential order of the reactions and consequences in the cellular redox equilibrium is displayed in Figure 6. β-Lapachone (ARQ761 in clinical form) undergoes a futile NQO1-mediated redox cycle, where the drug forms an unstable hydroquinone that spontaneously regenerates to the parent compound in a two-step oxygenation process [44,80]. The futile recycling produces elevated superoxide levels and eventually high hydrogen peroxide (H_2_O_2_) concentrations that lead to profound DNA damage and endoplasmic reticulum Ca^2+^ release. These events result in the DNA repair enzyme, poly(ADP-ribose) polymerase 1 (PARP1), being overwhelmed with the large levels of DNA damage, and the elevated nuclear Ca^2+^ levels enforce PARP1 hyperactivation [81]. Consistent with these data, the presence of catalase, which breaks down hydrogen peroxide, has been shown to downregulate the β-lapachone activation of PARP1 and subsequent cell death [112]. The formation of branched poly(ADP-ribose) chains results in a dramatic loss of NAD and ATP contents, thereby downregulating the cellular metabolism and energetics on the way to apoptosis. Cell death appears to occur independent of oncogene driver- or passenger-gene mutations but with a direct relation to the NQO1 content in the cells [81]. Inhibition of GAPDH and overall prolonged suppression of glycolysis, as marked by decreased glucose utilization and lactate [113], have also been noted.

The various routes leading to cell death in β-lapachone-treated cells are shown in Figure 7. One of them is the oxidation of reactive thiol groups in the mitochondrial potential transition pore complex to sulphenic acids [114], culminating in its functional inactivation, cytochrome c release, and apoptosome formation. Second is calcium release and the generation of endoplasmic reticulum stress. Thirdly, the involvement of topoisomerases in the production of single- or double-stranded DNA breaks has been implicated [115,116]. Many of these events, either singly or in combination, may enact apoptosis or necroptosis, in which the calpain protease is a major player [117].

### 2.10. Clinical Advances of β-Lapachone

Starting in 2003, tumor regression in several xenograft studies prompted the possibility of the clinical use of β-lapachone, particularly for NQO1-positive cancers [72]. Formulations of β-lapachone prepared in hydroxypropyl-β-cyclodextrin for improved solubility and ARQ 501 (ArQule, Woburn, MA, USA), a precursor analog, were used in a small study of 42 patients with different cancers in a phase 1 clinical trial completed in 2018 [76]. Despite some evidence of stable disease, the most common adverse effect was anemia (methemoglobinemia) in an overwhelming 79% of patients [69]. Since then, nanoparticle formulations of β-lapachone have been devised, and intratumoral injections given in experimental settings [73]. These studies highlight insolubility, toxicity, and hindrance to structural modifications as some problems that have reduced the therapeutic value of β-lapachone.

Deoxynyboquinone (DNQ) is another fairly well characterized natural product quinone substrate participating in futile catalysis by NQO1 [83,84]. Figure 8 depicts the structure, redox cycling, and cellular responses observed after DNQ exposure. Studies showed that DNQ was more potent than β-lapachone as an NQO1 substrate and that cells rapidly generated ROS after DNQ exposure [83,118]. NQO1-mediated reduction, followed by spontaneous reoxidation to the quinone, was proposed to be the mechanism of ROS production (Figure 8). The generation of sixty moles of ROS per mole of DNQ was estimated. Hypoxia or the administration of antioxidants reduced the extent of cell death, indicating that the cytotoxic mechanism of DNQ relies on the generation of ROS. Additionally, changes in cell cycle progression and altered gene expression, with an increased production of enzymes involved in glutathione (GSH) synthesis, were noted [83,84]. Although a moderate antitumor activity similar to β-lapachone was observed, it is clear that solubility, toxicity, and difficulty synthesizing more potent analogs further hamper the clinical development of DNQ.

## 3. Overview and Brief Descriptions of NQO1-Activatable Fluorescent Probes for Imaging Cancer Cells/Tissues

Sensitive activity stains for enzymes selectively expressed in human cancers offer valuable tools for imaging, with wide applications in experimental, diagnostic, and therapeutic settings. The real-time high-resolution visual information gained in a spatio-temporal context in cells is invaluable and far superior to the semi-quantitative immunohistochemical detection of denatured proteins. Nontoxic fluorescence probes, the ones that are cell-permeable and activated by enzymatic catalysis to trigger the generation of fluorescence in cells, are widely sought in oncology [119]. Thus, near-infrared and infrared (NIR/IR) fluorescence probes have been developed for the detection of cancer biomarkers, including Ɣ-glutamyltransferase [120], β-galactosidase [121], nitroreductase [122], N-aminopeptidase [123], alkaline phosphatase [124], azoreductase [125], tyrosinase [126] and cathepsins [127]. The lack of absorption by or non-reactivity of NIR and IR fluorescence by cellular macromolecules (nucleic acids, proteins, membrane components), and their ability to penetrate deep into tissues without generating background fluorescence, are hugely advantageous for imaging, thereby making them a preferred choice [128]. The different modalities and applications of NIR technology have been extensively reviewed [129,130]. Briefly, the medical applications of NIR probes include hypoxia detection in tumors, the use of small ligands for protease activity, photodynamic therapies, and fluorescence-guided cancer surgery. Worthy of mention in this context is Pegulicianine (Lumisight), an optical imaging agent approved in 2024 by the Federal Drug Administration (FDA, Bethesda, MD, USA) for detecting residual cancer after breast-conserving surgery [131]. Pegulicianine is a peptide prodrug linked with a fluorescent agent that is optically inactive, but generates a fluorescent signal after its peptide chain is cleaved by cathepsins and matrix metalloproteases (MMPs) which are expressed at higher levels in breast tumors than in normal cells [132].

### 3.1. General Design and Approaches of Molecular Probes for NQO1 Sensing, Quantitation, and Imaging

Given the fact that NQO1 is similarly overexpressed in most solid tumors of breast [V64] [133], lung [134], prostate [135] colon [136], pancreatic [100], and brain cancers [137], and the prognostic and therapeutic importance of NQO1 in oncology, probes to detect its cellular activity have been widely explored [138,139]. Various NQO1-imaging modalities in the entire fluorescence spectral range including the near-infrared, infrared, and chemiluminescence, and NQO1-driven prodrug delivery approaches have been engineered. A large number of these compounds linked to dissimilar fluorophores have been designed. Many excellent reviews on NQO1-targeted imaging have been published [130,140,141,142,143]. While we have not listed every NQO1-targeted compound reported, the subsequent sections describe the major and representative agents that yield fluorescence groups following NQO1 catalysis. Furthermore, anticancer drugs such as camptothecin, podophyllotoxin, and 5-fluorouracil have been linked with NQO1 recognition groups for in situ enzyme-directed delivery [88,96,140] as shown in Figure 9. A theranostic approach that combines the delivery of antitumor drugs and diagnostic imaging through NQO1-directed probes that release near-infrared or infrared fluorophores in human tumors has been proposed and prototype compounds synthesized; some of these were reported to induce immunogenic cell death [144]. In this potential clinical application, one can envisage a cancer patient obtaining an intravenous injection of the NQO1-targeted fluorescence agent just before surgery. The surgeon, then using a fluorescence visualization system (Figure 9), can have two options: one, to differentiate the margins of malignant tissue from the adjacent normal tissue before surgery, and the second, a post-operative visualization to inspect the residual malignant tissue remaining and attempt a subsequent resection. In fact, the latter visualization method has been approved by the FDA for breast cancer surgery using Pegulicianine in the USA [131,132].

The NQO1 turn-on probes remain latent/quenched and non-fluorescent until the precise enzymatic cleavage when the fluorogenic group is released in cells. Structurally, the fluorescent probes usually contain an NQO1 recognition group, a connecting spacer, and a fluorophore (Figure 9). In 2012, McCarley and colleagues examined the factors associated with the enzymatic reduction of a group of substituted quinone propionic acids and NQO1. Their studies highlighted that trimethyl lock quinone propionic acid could serve as an optimal trigger group for NQO1 catalysis, highlighting its effectiveness as a recognition substrate aiding in the formation of intramolecular lactonization to activate reporter signals [145]. The trimethyl lock quinone group, due to its electron-poor nature, efficiently suppresses the fluorescence signal of the reporter component. This property serves as a key strategy in crafting activatable NQO1 probes (Figure 9). Some NQO1-activated probes are engineered by incorporating self-immolative linkers (such as the N-methyl-p-aminobenzyl alcohol (NMPABA)) between the trimethyl lock quinone and the reporter moiety. Here, the enzymatic reduction by NQO1 initiates the removal of the quinone substrate, triggering the breakdown of the self-immolative linker from the probe and subsequently releasing the free signal reporter component, as shown in Figure 9. Compared to conventional activatable NQO1 probes, those with self-immolative linkers also enhance the probe’s stability and augment the interaction between substrates and NQO1, thereby improving their reactivity.

### 3.2. NQO1 Fluorescent Probes with Emission Wavelengths in the Visible Spectrum

NQO1 activity imaging compounds can be categorized as either visible-range or near-infrared (NIR) probes, based on the fluorescent reporter unit variations in emission wavelengths. The structures and characteristics of prominent NQO1 fluorescent probes and their reported use in live cells and in vivo are covered briefly in the following sections. Our list is by no means exhaustive, and readers are referred to several excellent reviews for details of other NQO1 activity probes (Table 2).

The probe numbers in the second column refer to the original numbers used by authors in their publications. The chemical groups in different colors in the compound structures correspond to NQO1 recognition moieties, fluorophores, or the drugs conjugated.

Reagents 1 and 2 in Table 2, namely, the NMPABA and Q3N1, containing a trimethyl-locked quinone propionic acid trigger group attached to fluorescently masked naphthalimide through a self-immolative linker, was synthesized by McCarley’s group [138,146]. Cleavage of the probe by NQO1 resulted in green fluorescence with a λmax of 470 nm. Using confocal fluorescence microscopy, the authors imaged NQO1-positive HT29 cells from negative H596 tumor cells successfully with a positive-to-negative ratio of 500:1 [146]. Because Q3N1’s excitation wavelength is in the UV spectrum, the probe is not useful for tissue imaging.

In the year 2015, the same research group introduced an even more sensitive NQO1 probe that quickly became active in the presence of NQO1 [147]. It was excellent at distinguishing between NQO1-proficient and -deficient cells (reagents 3 and 4). These probes had a high positive-to-negative fluorescence intensity ratio of around 135-fold. This high ratio happened because the NQO1 substrate, quinone propionic acid, was near the part that blocks the fluorophore, causing it to quench. This was driven by the way energy moved in the probe structure [148,158].

Again in the year 2015, McCarley’s laboratory. synthesized a probe using a rhodamine dye linked with quinone propionic acid [148]. This process involved two stages: first, when NQO1 was present, the trigger group Q3PA changed, forming a specific hydroquinone. Then, this intermediate product broke down by itself, releasing the lactone and the glowing part of the dye. In the same year, they improved the previous rhodamine probe (reagent 6) and also prepared and characterized several rhodamine analogs [149]. Among these analogs, Q_3_MJSNR stood out as an excellent stain because of its stability for 39 h and bright optical properties.

Next, Fei et al. [150] created a nanoprobe by using small molecular arrays prepared from 6-hydroxyphenyl-BODIPY and assembling them into micelles (reagent 7). This nanoprobe was effective in distinguishing NQO1 activity levels in cancer cell lines. Subsequently, Cuff et al. [151] used the properties of various dyes and chose 4-methylumbelliferone to design an NQO1-activated probe (4-MU; reagent 8). This fluor was able io image enzyme activity within 10 min. SYZ-30, the next reagent, number 9, is based on the 7-nitro-2,1,3-benzoxadiazole (NBD) fluorophore. SYZ-30 quickly responded to detect NQO1 activity in tumor cells within 5 min and was suitable for imaging NQO1-positive cancer cells [152].

In 2017, Kwon et al. developed reagent 10, aimed at detecting NQO1 using two-photon excitation [153]. This probe combined an aminoacetyl-naphthalene motif with a trimethyl lock quinone. Probe 10 responded rapidly and specifically to NQO1, showing an 8-fold increase in fluorescence intensity in 4 min. It was effective in monitoring NQO1 in cancer cells using two-photon excitation and demonstrated high photo stability. Additionally, Jiang and Kima also developed several TP probes to distinguish between cancer cells and normal cells based on their NQO1 expression levels [159].

In 2020, Zhou’s team created a special two-photon probe called reagent 11 (QBMP) to study NQO1 in mitochondria and cells [154]. They synthesized QBMP by joining hydroxylphenylpolyenyl pyridinium fluorophore with trimethyl lock quinone. On testing, QBMP’s green light changed to red on excitation at 407 nm and emission at 566 nm, respectively. QBMP did well in tests; it responded within 4 min and was used to measure NQO1 levels in cancer cells and a Parkinson’s disease model and preserved brain tissue at a depth of 225 µm. Because of these properties, the probe has clinical promise.

In 2016, Kim and colleagues developed a probe for both therapy and diagnostics that targeted the mitochondria [157]. They engineered reagent 12 by putting together a triphenylphosphonium group which targets mitochondria, with an AIE (Aggregation-Induced Emission) generator called tetraphenylethene, all connected to an NQO1-responsive quinone group. When they directed this probe to the mitochondria in cancer cells, the AIE dye was released by NQO1 catalysis, which subsequently formed aggregates and perturbed the mitochondrial metabolism, leading to cell death. The background fluorescence of this reagent was low, and there was a 500-fold increase in signal after NQO1-triggered release. However, the probe was not useful for in vivo applications, given its excitation in the ultraviolet spectrum.

Another approach was to synthesize prodrugs that could be activated by NQO1 and the cytotoxic moieties released preferentially in cancer cells. In 2016, Kim’s group designed reagent 13 by combining the topoisomerase I drug 7-ethyl-10-hydroxycamptothecin (SN-38) with hydroquinone (activated by NQO1) and biotin for targeting cancer cells [155]. In the presence of NQO1, the fluorescence signal became stronger in 30 min at 550 nm, suggesting that SN-38 had been released. Also, SN-38 in this setting was tumor-selective, influenced dually by the extent of biotin receptor distribution and NQO1 abundance. Thus, the cancer cells A549 and HeLa were killed selectively because of higher NQO1 expression and increased biotin binding to the cell surface [160].

Punganuru et al. [156] from our laboratory characterized an NQ-DCP fluorescent probe (reagent 14 in Table 2) for determining the NQO1 activity in cancer cells. The NQ-DCP probe had a wide difference in light color after emission (145 nm), was cell-permeable and non-toxic, and detected NQO1 activity quickly in brain tumor cells. NQ-DCP was made by combining QPA and a glowing part called DCP with an ester linkage introducing some non-specificity.

### 3.3. NIR/IR Probes for NQO1 Activity Imaging

The visible spectrum-range NQO1 activity probes suffer from having low signal/background fluorescence ratios and an inability to penetrate deeper into tissues [161]. To tackle these issues, NIR (near-infrared) probes for NQO1 have been preferred. These probes have longer-wavelength fluorescence groups, allowing NQO1 activity visualization and quantitation in human malignant tissues [162,163]. In 2016, McCarley’s team prepared a probe called Q3STCy (reagent 15 in Table 3), which emitted in the NIR range following cleavage by NQO1 and the release of tricarbocyanine (TCy) fluorescence reporter [164]. The large Stokes shift of 149 nm was highly beneficial in distinguishing the tumor cells from their normal counterparts based on NQO1 content. For example, they observed a bright red fluorescence in colorectal carcinoma cells (HT-29) and ovarian cancer cells (OVCAR-3) which are proficient in NQO1; however, for non-small cell lung carcinoma cells (H596) deficient in NQO1, there were no signals. Additionally, this probe imaged the NQO1 activity in three-dimensional colorectal tumor models and metastases in a mouse model of ovarian cancer [161].

In our laboratory, Punganuru et al. in 2019 synthesized and characterized an NIR NQO1 turn-on probe designated as NIR-ASM (reagent 16), which is a nontoxic highly effective non-invasive imaging agent for in vitro and in vivo analyses [165]. The applications of NIR-ASM in cell culture and live tumor xenografts have been described in Figure 10, Figure 11 and Figure 12 of this review.

Next, Shen and colleagues in 2017 prepared yet another sensitive NIR probe called HCYSN (reagent 17) for NQO1 activity. It had an emission maximum of 695 nm and was able to detect tiny amounts of NQO1 (as low as 4.9 ng/mL or 0.49 mU/mL) in cancer cells [157]. In 2021, Zhang’s team created an NIR probe for NQO1 using naphthoquinone and methylene blue groups. This probe was highly sensitive in detecting NQO1 both in lab tests and A549 tumor xenografts, with an emission at 695 nm. The authors also reported a slight reduction in tumor volume in the xenografts, making this a theranostic probe [139].

In 2022, Lin et al. reported a hemi-cyanine dye-based probe called LET-10 (reagent 18) for both near-infrared (NIR) fluorescence and photoacoustic imaging to visualize NQO1 activity [169]. The LET-10 probe, in response to NQO1, could also adjust its own signal for clearer imaging by using naphthalocyanine as a built-in guide. They conjugated the LET-10 probe with a Q3PA arm. In the presence of NQO1, the dye was released to generate both NIR fluorescence (at 730 nm) and photoacoustic signals (at 700 to 740 nm) for dual imaging. Furthermore, the probe was encapsulated in a PEG formulation to make the two signals mutually inclusive [169].

In 2019, Zhao et al. devised an innovative NQO1 activity probe whose emission could be controlled at NIR-I (650–900 nm) or NIR-II (1000–1700 nm) with large Stokes shifts [168]. These smart probes were created by linking BODIPY dyes to NQO1 substrates with self-immolative linkers in between. One of these probes, NQO-ImI (reagent 19), had a two-channel ratiometric response, with the original 557 nm absorption band changing to a 675 nm band in the presence of NQO1 [168] (Table 3).

### 3.4. Chemiluminescent Probes for NQO1

Chemiluminescence (CL) offers significant benefits compared to routine fluorescence, such as reduced background interference, deeper penetration, and more sensitivity [170,171,172,173,174]. In 2019, Song’s team prepared a CL probe that could detect NQO1 in cells using near-infrared light. This probe, known as CL-P (reagent 20), shined bright at 725 nm after NQO1-mediated cleavage [175]. NQO1 at small levels (0.134 µg/mL) was visualized within 10 min. Cells treated with dicoumarol showed diminished chemiluminescence, verifying the NQO1 specificity [175].

In 2008, Kim and his team reported a CL probe incorporating a trimethyl-locked quinone trigger moiety covalently tethered to phenoxy-dioxetane through a self-destructing linker (reagent 21 in Table 4). The probe was analyzed in tumor cell cultures and an A549 lung cancer animal model. It was cell-permeable, safe, and capable of producing a “turn-on” chemiluminescence response in NQO1-positive tumors [176].

### 3.5. Bioluminescent Probes for NQO1 and NQO1-Deliverable Anticancer Drug Strategy

Some organisms have an impressive ability to make light from chemical energy, and this has inspired a whole area of study called bioluminescence imaging (BLI) [179,180]. BLI has great promise for imaging human tumors [181]. The advantages include little background, deeper tissue penetration, and strong signals in the images [182,183,184]. Bioluminescence involves the enzyme luciferase, and the substrates luciferin and oxygen. ATP and Mg2+ are other reaction components [185]. In 2013, Zhou and colleagues synthesized many trimethyl lock quinone luciferin molecules [186] and used them as bioluminescent substrates for NQO1. The luciferin-linked reagents 22 and 23 in Table 4 were synthesized by adding N, N’-dimethylethylenediamine spacers to reduce the background bioluminescence [186]. The luciferin released after NQO1 cleavage was used to measure NAD(P)H in biological samples and monitor cell viability via NAD(P)H-dependent cellular oxidoreductase enzymes and their NAD(P)H cofactors.

The tumor abundance of NQO1 has also been exploited to deliver anticancer drugs like SN-38, an active topoisomerase I (topo I) drug. Irinotecan, a prodrug by itself, is cleaved by carboxylesterases in cells to release SN-38, which induces topo I-associated DNA breaks. Instead of irinotecan, an alternative route was taken to incorporate the SN-38 structure (reagent 24) through modifications at carbon 10 [187]. Targeting the tumor microenvironment, an NQO1-SN38 prodrug has been described, along with a lysyl-oxidase inhibitor [188]. While yet in their infancy, such drug discovery approaches appear promising.

Phenalenone is an efficient synthetic photosensitizer with a near-unity singlet oxygen quantum yield. Digby et al. in 2020 [160] exploited NQO1 as an optical therapeutic. They prepared a probe using phenalenone, which is initially quenched via photo-induced electron transfer by the attached quinone (reagent 25). The native phenalenone is liberated in the presence of NQO1, resulting in the production of cytotoxic singlet oxygen upon irradiation. NQO1-mediated activation of this probe in A549 lung cancer cells induced a dose-dependent photo-cytotoxic response after irradiation. In contrast, no photo-induced cytotoxicity was observed in the MRC9 normal lung cell line.

In a theranostic approach for simultaneous drug delivery and cell imaging driven by NQO1, Li et al. in 2017 [161] designed a prodrug containing two camptothecin (CPT) moieties as the anticancer drug, a DT-diaphorase-responsive quinone propionic acid moiety, and a set of self-immolative linkers (reagent 26). The presence of NQO1 in cells led to the release of two CPT molecules and the restoration of the fluorescence of the latter, thus allowing the fluorescence-mediated monitoring of NQO1 levels, as well as the tracking of CPT release. Upon internalization by NQO1-proficient cells, the prodrug released fluorescent CPT and elicited a potent cytotoxicity (IC_50_ of 0.71 μM) against cancer cells.

### 3.6. Biomedical Applications of NIR-ASM, a Superior Nontoxic NQO1-Imaging Agent

In 2019, our research group [165] designed, synthesized, and characterized a stable NIR fluorescent probe for NQO1 called NIR-ASM [reagent 16 in Table 3, ref. [168]. NIR-ASM appears to be the most ideal currently available for the non-invasive imaging of NQO1 live animals in cultured cells. The NQO1 substrate quinone propionic acid (QPA) was linked with dicyanoisophorone (ASM) as the fluorophore with an amide spacer (instead of an ester) for conferring stability and specificity (Figure 10A). The probe turns on after responding to NQO1 in malignant tissues and is now commercially available from Millipore-Sigma Company, Rockville, MD, USA [189]. Compared to other probes for NQO1, NIR-ASM has a large Stokes shift (~186 nm), and its maximum emission wavelength is at 646 nm, with an intense red fluorescence following NQO1-specific release [165]. Figure 11 is a diagrammatic display of the different applications of NIR-ASM in cell culture and live nude mice bearing human xenografts and the results expected in NQO1-positive and -negative backgrounds. We used several NQO1-proficient and -deficient human tumor cells for characterizing the probe. Confocal fluorescence analysis showed that only the cancer cells (but not the IMR-90 and HUVEC normal cells) responded with bright red fluorescence proportional to the higher abundance of NQO1 in A549 cells and the lower levels in H460 cells (Figure 12A). To explore the potential use and specificity of NIR-ASM for the real-time imaging of NQO1 activity in human cancers, we injected A549 (NQ01-positive) and MDAMB-231 (NQO1-negative) cells into nude mice and established subcutaneous xenografts. These live animals with tumors received tail vein injections of NIR-ASM to allow a swift distribution of the imaging agent in the circulation. The imaging of the animals immediately after injections using an in vivo imaging system revealed the rapid generation of a fluorescence signal and its accumulation over time only in A549 tumors but not in the NQ01-absent MDA-MB 231 tumors, even when both tumor masses were present in the same animal (Figure 12B). Fluorescence signals were observed only in the tumor and not in any other major organs such as the lung, hear, spleen, kidney, intestine, and liver. These studies reveal the potential utility of NIR-ASM for clinical applications.

## 4. Conclusions

We reviewed expression patterns, bioreductive futile substrates, and various fluorescent probes to image and quantify NQO1 activity in human cancers. It is evident that NQOs, comprising NQO1 and NQO2, which catalyze the two-electron reduction of endogenous and exogenous quinones, are complex and multifaceted enzyme systems, of which the former has been studied the most. The inducibility of *NQOs* by oxidative stress, the presence of functionally inactivating gene polymorphisms, particularly for *NQO1,* the yet undefined functions of NQO2, and how these enzymes interact in the cell biology of quinone reduction, all add to the difficulty of estimating the active enzyme content in cells and ascribing specific functions. Some reports do indicate that NQO2 and NQO1 share common substrates such as acetaminophen, aripiprazole, RH1, and others [190]. Therefore, it is not surprising that the activation of mitomycin and other anticancer quinones has been shown to occur by many enzymes, including NQO1 [91,92]. A majority of the quenched fluorescence probes described in this review that release powerful intracellular fluorophores following NQO1-driven cleavage also suffer from similar deficiencies; they have failed to demonstrate enzymatic specificity. There is a need for validating the probes in vitro in the presence of purified NQO1 protein and/or silencing the *NQO1* expression and demonstrating the lack of fluorescence production. Indeed, our studies applied both of these methods in characterizing NIR-ASM as a specific probe for NQO1 [165].

Because of variations in NQO1 levels in normal and cancer tissues, careful consideration in preclinical studies should be given to address drug efficacy and toxicity. This cautionary note also justifies the use of imaging technology in biopsy specimens to visualize the enzyme abundance and select NQO1- or non-NQO1-activated drugs accordingly. Despite the concerns and limitations noted, there is no doubt that NQO1 has evoked great interest encompassing the areas of tumor biology, drug discovery, chemistry, and cancer imaging and has emerged as a tumor-selective drug target. Continued research and innovative technologies in the coming years are likely to assist in designing potent NQO1 futile substrates suitable for the treatment of different human cancers, NQO1-based reagents for drug delivery, guided cancer surgery, and combined theranostic compounds that deliver the drug payload and also allow tumor imaging. 

## Figures and Tables

**Figure 1 cells-13-01272-f001:**
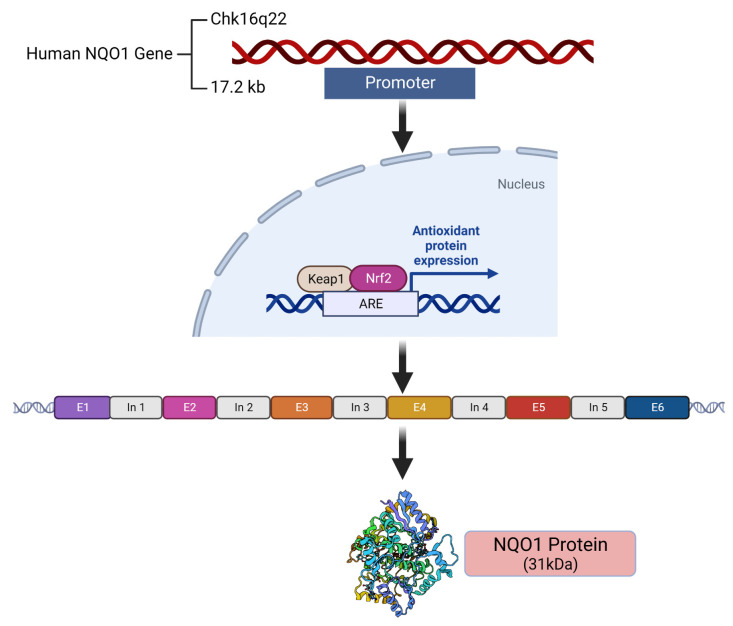
The location and composition of the human *NQO1* gene and its expression controlled by the ARE-NRF2 pathway are shown. E = exon, In = intron.

**Figure 2 cells-13-01272-f002:**
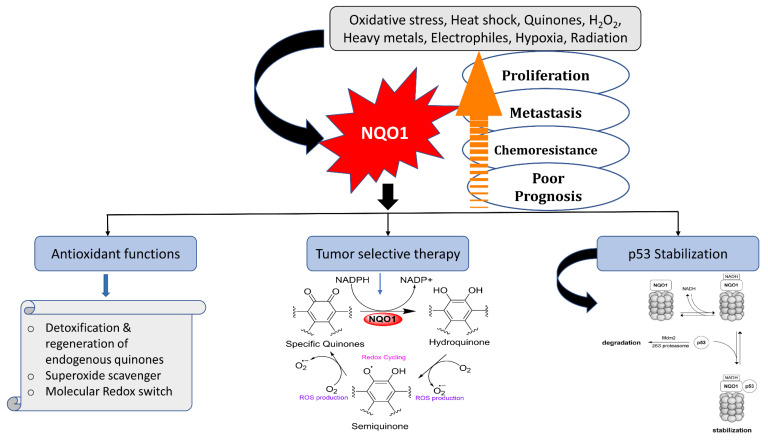
Functions and roles of NQO1 in quinone metabolism and cellular defense against oxidative stress and other stresses, and impact on p53 steady-state levels are represented. Linkage of increased NQO1 expression with carcinogenesis and opportunities for selective therapeutic strategies through futile substrates is displayed.

**Figure 3 cells-13-01272-f003:**
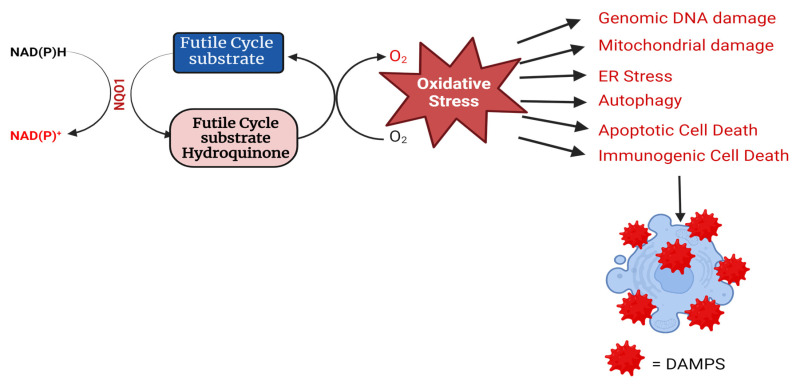
Catalysis of a futile cycle substrate by NQO1 and the linkage of the resulting redox imbalance to different routes of cell death are shown. ER, endoplasmic reticulum; DAMPS, damage-associated molecular patterns.

**Figure 4 cells-13-01272-f004:**
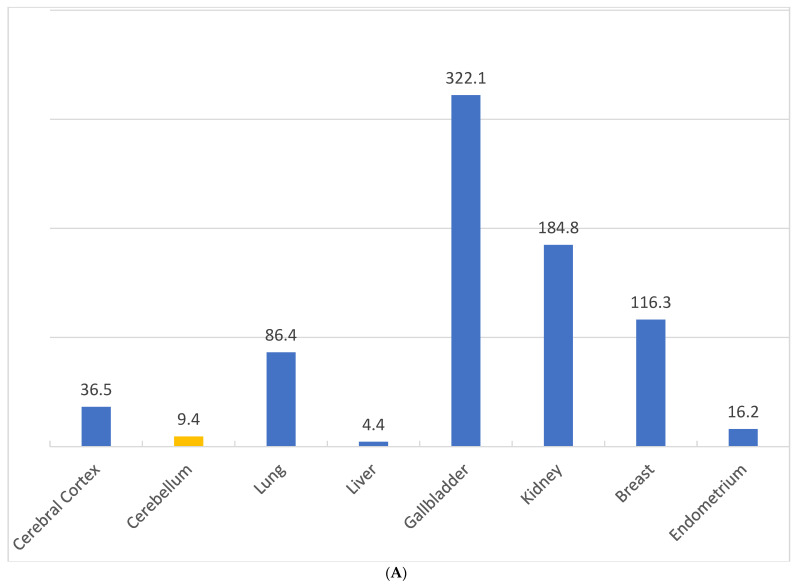
(**A**). *NQO1* FANTOM Tag expression as reported in different human tissues. The tissue data for RNA expression obtained through a Cap Analysis of Gene Expression (CAGE) generated by the FANTOM5 project are shown. The values represent scaled Tags Per Million (data adapted from https://www.proteinatlas.org/ENSG00000181019-NQO1/tissue, accessed on 27 July 2024). (**B**). Relative NQO1 protein expression in different human cancer types. Most cancers display strong cytoplasmic positivity in a fraction of cells, greater in lung cancer, followed by colorectal, endometrial, stomach, and pancreatic cancers (Data adapted from https://www.proteinatlas.org/ENSG00000181019-NQO1, accessed on 24 July 2024).

**Figure 5 cells-13-01272-f005:**
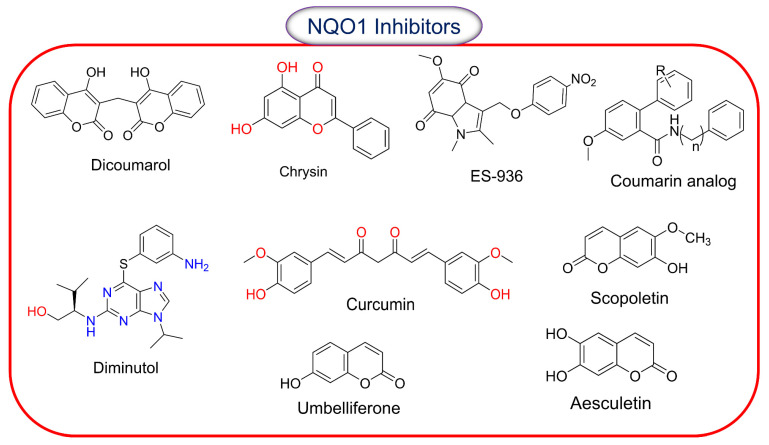
Chemical structures of compounds reported to inhibit the catalytic activity of NQO1.

**Figure 6 cells-13-01272-f006:**
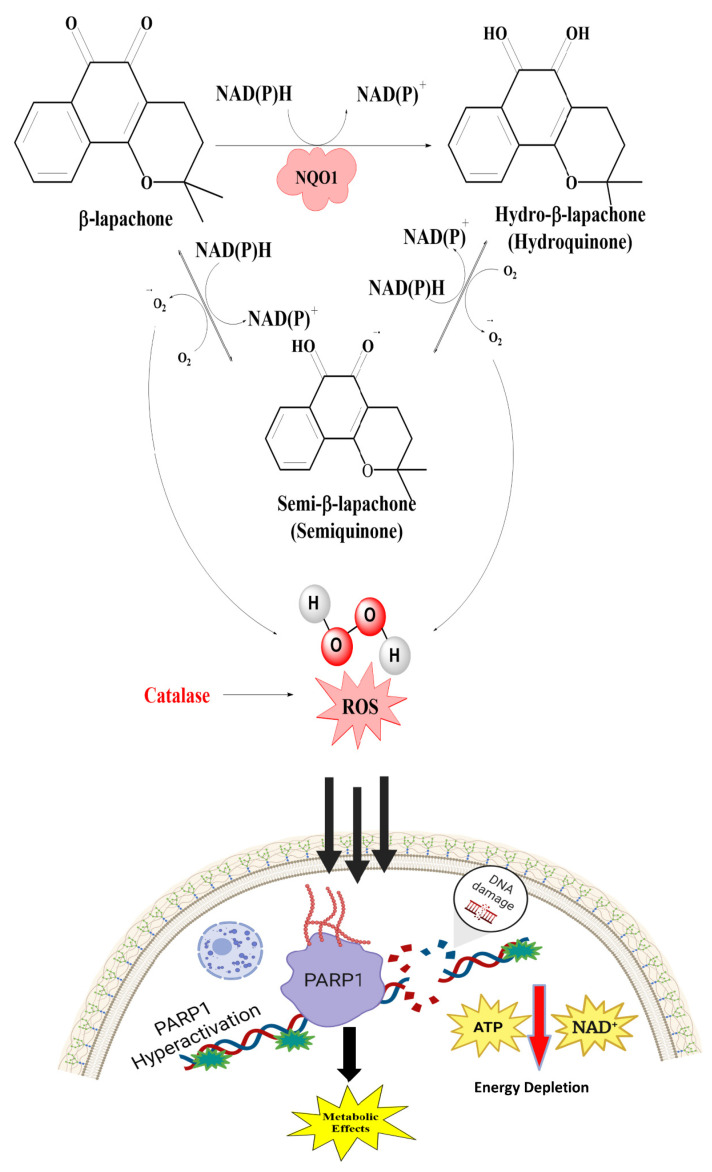
The mechanism of action of β-lapachone. The futile catalysis of the substrate to its hydroxyquinone and semiquinone forms and the consequent generation of ROS, increased production of H_2_O_2_, consequent PARP-1 activation, and metabolic perturbations are shown.

**Figure 7 cells-13-01272-f007:**
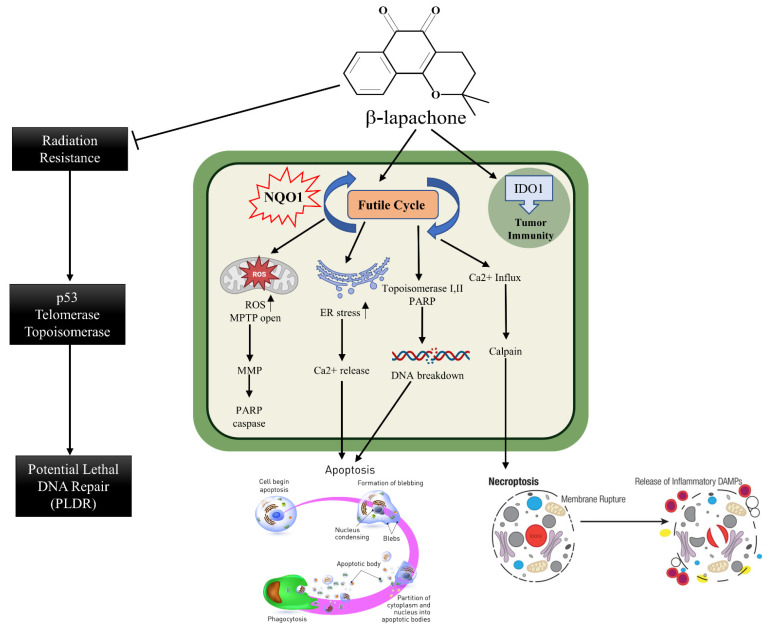
Details of cellular mechanisms leading to cell death, necroptosis, and release of DAMPS by β-lapachone are represented. Inhibition of radiation resistance and DNA repair as reported in the literature are also shown.

**Figure 8 cells-13-01272-f008:**
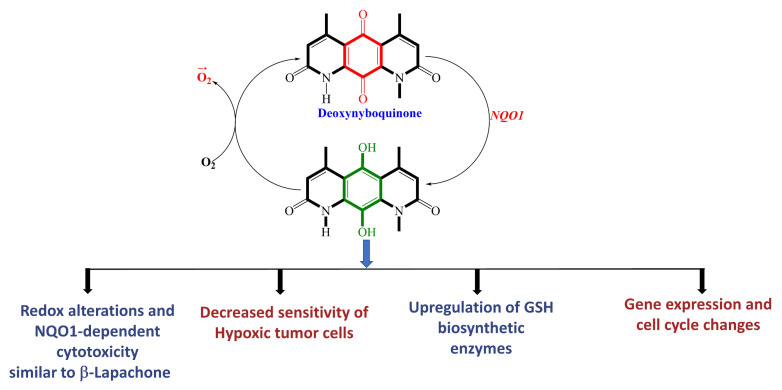
Futile catalysis of deoxynyboquinone by NQO1 and the resulting changes in cellular redox and physiology are shown.

**Figure 9 cells-13-01272-f009:**
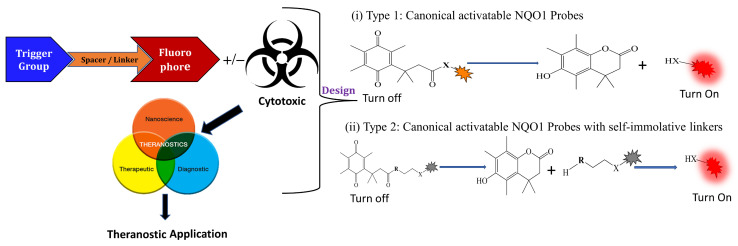
Design strategies of NIR fluorescent probes for detection and quantitation of NQO1 activity in cells and theranostic applications thereof in human cancers. A general design and sensing mechanism of canonical activatable NQO1 probes and self-immolative linker-based activatable NQO1 probes is shown.

**Figure 10 cells-13-01272-f010:**
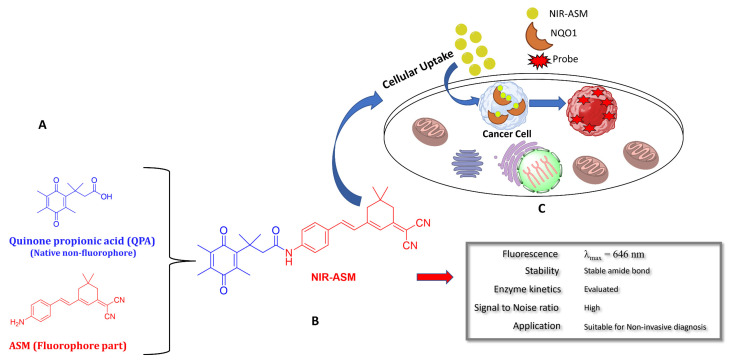
The structure of an NQO1-activatable NIR fluorescent probe called NIR-ASM (shown in part (**B**)). (**A**) Synthesis of NIR-ASM by coupling ASM with quinone propionic acid (QPA) and properties of NIR-ASM. (**C**) NIR-ASM activation and fluorescence generation in NQO1expressing cancer cells.

**Figure 11 cells-13-01272-f011:**
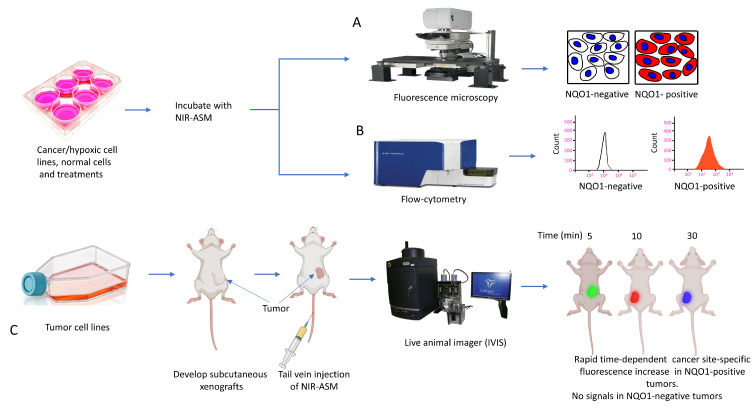
(**A**) Diagrammatic summary of techniques for NQO1 imaging using NIR-ASM and resulting fluorescence in NQO1-positive cancer cells and lack of it in NQO1-negative (normal) cells after incubation of 10 µM NIR-ASM for 1 h is shown. (**B**) Flow cytometry assays to determine NIR-ASM activation by NQO1-positive and -negative cell lines (**C**) Application of live in vivo fluorescence imaging of NQO1-positive tumors developed in nude mice.

**Figure 12 cells-13-01272-f012:**
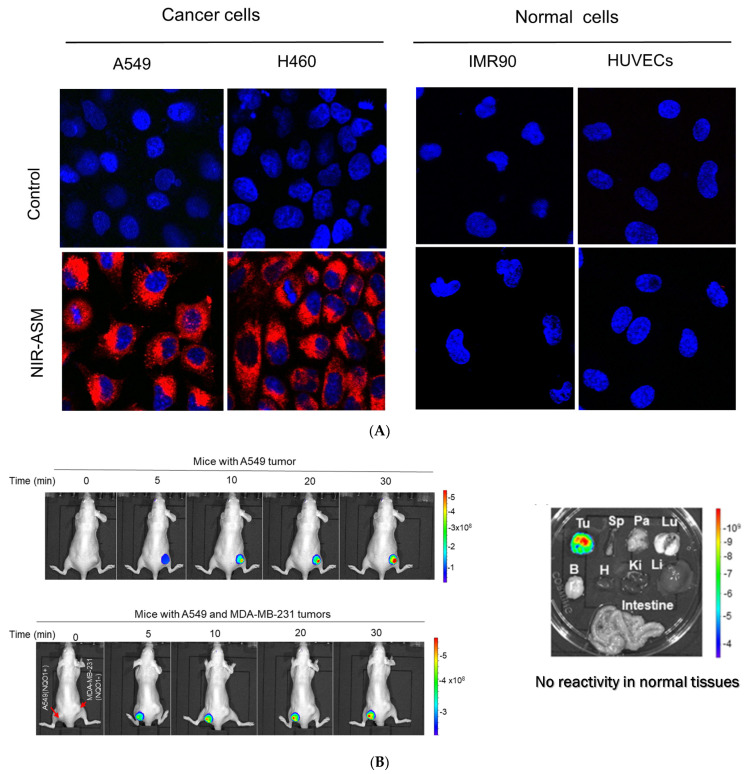
(**A**): Confocal red fluorescence images of NQO1-positive cancer cells and NQO1-negative normal cells after incubation of 10 µM NIR-ASM for 1 h. The cell nuclei were counterstained with Hoechst 33342 (reproduced from ref. [168] with permission). (**B**): Results of live animal fluorescence imaging of endogenous NQO1 activity in A549 tumor xenografts after intravenous administration of NIR-ASM (5 mg/kg) to tumor-bearing nude mice (reproduced from ref. [168] with permission). The upper panel shows mice with NQO1-proficient A549 tumors and the lower panel shows the same animal with both A549 and NQO1-deficient MDA-MB-231 tumors on the left and right flanks respectively. Tumors and organs were harvested from the A549 tumor-bearing mice after NIR-ASM injections and imaged for ex vivo fluorescence as well. Fluorescence only in the tumor but not in the spleen (Sp), pancreas (Pa), lungs (Lu), brain (B), heart (H), kidneys (Ki), liver (Li), and intestine is evident, showing specificity.

**Table 1 cells-13-01272-t001:** NQO1-activated substrates/prodrugs.

Compound	Structure	Properties/Limitations to Therapy	Reference(s)
β-Lapachone	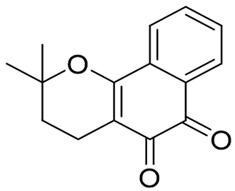	Well-characterized, excellent futile substrate, complex structure, solubility problems, potent efficacy in vitro and in preclinical cancer models; however, anemia was a prominent adverse effect in a single clinical trial.	[44,69,75,80,81,82]
Deoxynyboquinone	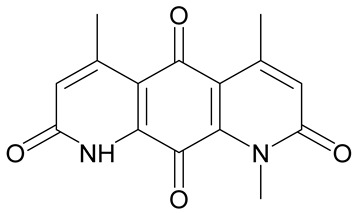	Natural product, has also been synthesized, more potent than β-lapachone as a futile cycle substrate and for induction of tumor cell killing, insoluble, limited data on anticancer efficacy.	[83,84]
GNQ-9	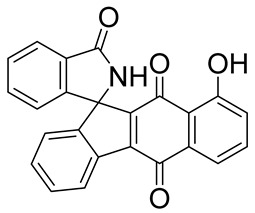	Synthetic quinone substrate, BBB-permeable, strong cytotoxicity in cancer cell lines, eliminated glioblastoma in orthotopic xenografts. Some evidence indicates immunogenic cell death may play a part.	[79]
Phenothiazinium redox cyclers	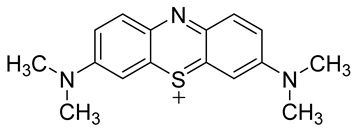	Robust induction of oxidative stress, and cell death in NQO1 stably transfected MCF-7 cells was demonstrated.	[85]
RH1	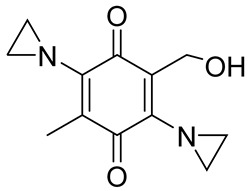	RH1 is similar to mitomycin, activation by NQO1 and similar enzymes leads to the formation of cytotoxic species, which then alkylate and crosslink the DNA. Phase I clinical trial indicated bone marrow suppression in patients.	[52,86]
Combretastatin A-4 prodrug	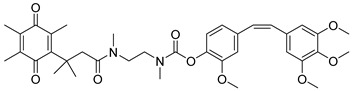	CA-4 is an anti-microtubule and anti-angiogenesis drug. NQO1-mediated release of CA-4 was reported in a prodrug approach. Selective killing of NQO1-rich cells was indicated.	[87]
Podophyllotoxin prodrug	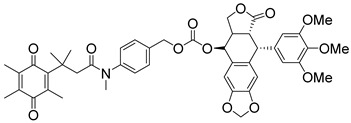	Podophyllotoxin derivatives inhibit topoisomerase II. Prodrug 3 of the study induced greater cell kill in NQO1-positive cells and suppressed the growth of HepG2 xenografts.	[88]
Steptonigrin (STN)	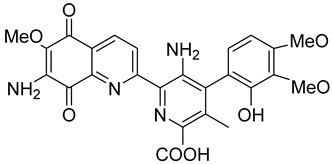	A quinone antibiotic, has been reported to induce NQO1-dependent and dicumarol-sensitive cell death.	[89,90]
Geldanamycin AAG	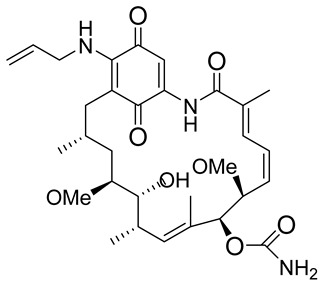	The hydroquinone resulting from NQO1 action on Gelda-AAG inactivates HSP-90, leading to apoptosis. Thus, these cytotoxic quinones exhibit synthetic lethality with NRF2 inducers.	[39]
Mitomycin C (MMC)	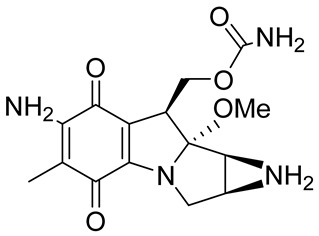	Mitomycin C (MMC), an established clinically used antitumor antibiotic, can be activated by NQO1 to alkylate DNA and generate crosslinks. Other enzymes have also been implicated in MMC conversion to semiquinone species.	[67,68,91,92]
EO9	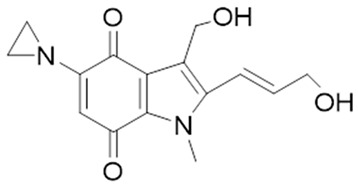	EO9 was engineered as an NQO1-targeted drug; however, it penetrates the cancers poorly and is rapidly eliminated, thus hindering its antitumor efficacy.	[66,93,94,95]
5 FU prodrug	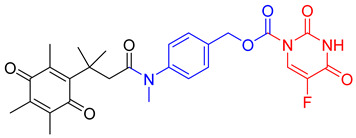	A tripartite prodrug composed of a trigger group, a self-immolative linker, and 5-FU was synthesized. The prodrug had a cytotoxicity similar to 5-FU and may have a better safety profile.	[96]

**Table 2 cells-13-01272-t002:** Designation and structures of reported NQO1 turn-on fluorescent probes.

Reagent Number	Original Designation	Structure	Reference
1	NMPABA	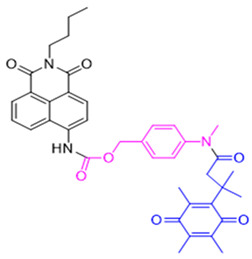	[138,146]
2	Q3NI	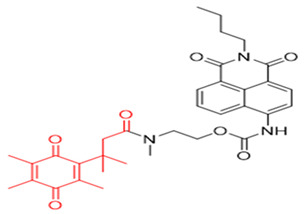	[146]
3	Q3PA	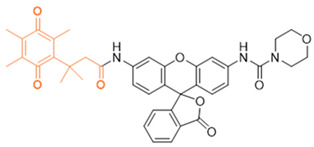	[146]
4	QMeNN	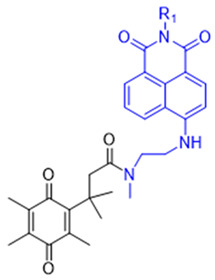	[147]
5	Q_3_PA Rhodamine fluorophore	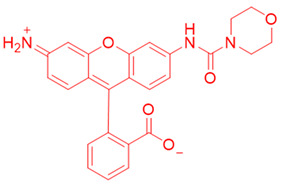	[148]
6	Q_3_MJSNR	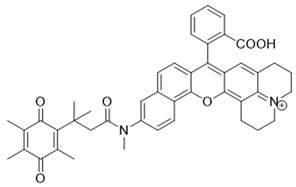	[149]
7	6-Hydroxyl phenolprobe-13	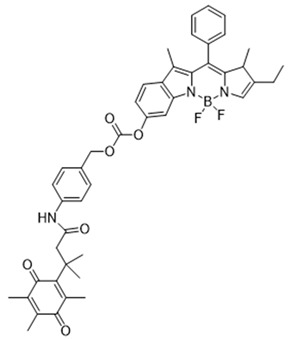	[150]
8	4-Methylumbelliferone (4-MU), probe-14	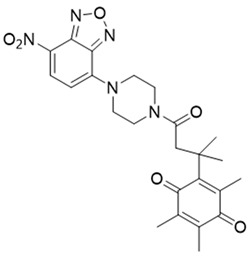	[151]
9	7-Nitro-2,1,3-benzoxadiazole, probe-16	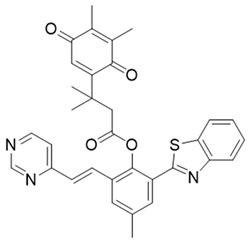	[152]
10	Aminoacetyl-naphthalene, probe,18-TPQ	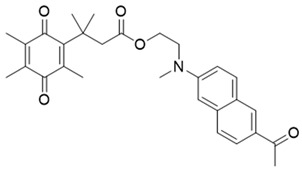	[153]
11	Hydroxylphenylpolyenyl pyridinium, probe 19-QBMP	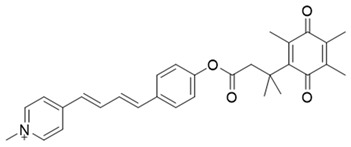	[154]
12	7-Ethyl-10-hydroxycamptothecinprobe-24	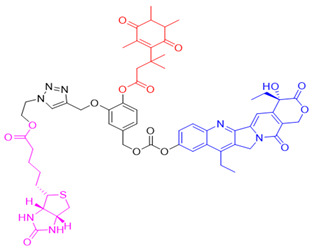	[155]
13	NQ-DCM	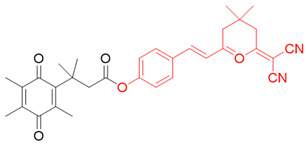	[156]
14	Triphenylphosphoniumprobe-21	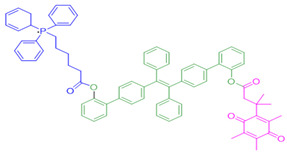	[157]

**Table 3 cells-13-01272-t003:** Designation and structures of quenched NIR probes turned on by NQO1 reported in literature.

Reagent Number	Designation	Structure	Reference
15	Q_3_STCY	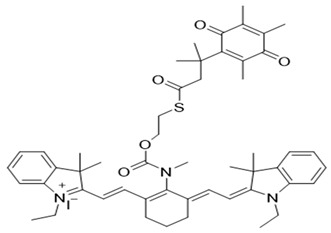	[164]
16	NIR-ASM	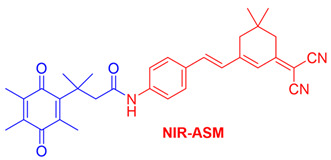	[165]
17	HCYSN	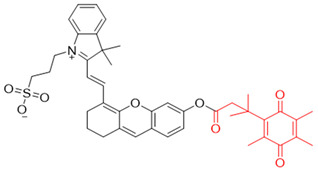	[166]
18	LET-10	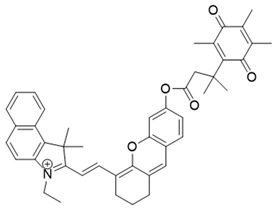	[167]
19	NQO-Iml	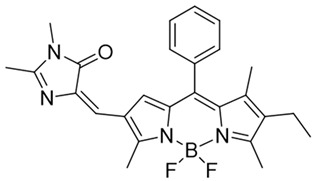	[168]

**Table 4 cells-13-01272-t004:** Miscellaneous probes activated by NQO1 to generate chemiluminescence, bioluminescence, and/or drug release.

Reagent Number	Chemiluminescent Probes for NQO1	Structure	Reference
20	NIR CL probe	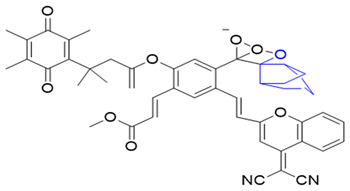	[175]
21	LET-10	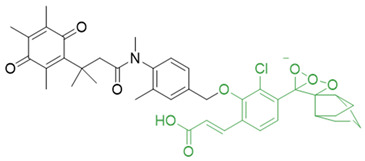	[176]
22	Luciferin probe 1 (Bioluminescent)	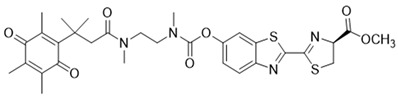	[177]
23	Luciferin probe 2 (Bioluminescent)	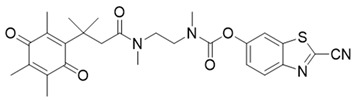	[177]
24	SN-38 conjugate	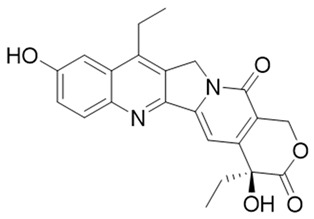	[178]
25	Phenalenone conjugate	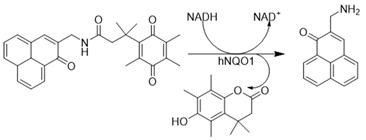	[165]
26	Prodrug for detecting DT-diaphorase	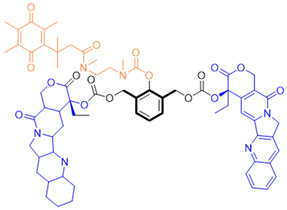	[161]

## Data Availability

Data are available from the authors upon request.

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
