# Peer review of "Human NQO1 as a Selective Target for Anticancer Therapeutics and Tumor Imaging"

_cells, 2024, doi:10.3390/cells13151272_

Round 1
Reviewer 1 Report
Comments and Suggestions for Authors
This review focuses on NQO1 and its clinical application in antitumor therapy and imaging. Although many of these areas have been reviewed previously the authors have brought together two major areas, therapeutics and imaging, which is the novelty of this manuscript. The underlying premise of the review rests on two central building blocks 1) that human tumor tissue contains increased levels of NQO1 relative to normal tissues and 2) that imaging probes can be selectively turned on in tumors after metabolism by NQO1. I thought the manuscript overall was well-written and presented. However, I have two major issues with the manuscript that need to be addressed and a number of minor issues.
Major points
1) While it is generally true that human cancers and in particular solid tumors have high levels of NQO1, this is not always the case and there are a number of significant exceptions. The authors own figure 4b makes this exact point. There are also many human tissues which do have significant and even elevated levels of NQO1 (see ref 41 for example). The authors should expand their discussion to address these points and raise the issue that toxicity to normal tissues from NQO1-directed antitumor agents needs to be carefully addressed during any pre-clinical studies. This is an important point which will likely limit the use and utility of NQO1-directed agents and needs to be discussed.
The point should also be made that mRNA profiling should not be relied upon to assess NQO1 levels because of the widespread NQO1*2 polymorphism which results in minimal NQO1 activity even when high levels of NQO1 mRNA are present. This is also a good argument for the use of NQO1 activated probes to assess NQO1 directly in tumors.
2) The authors have nicely summarized the use of NQO1 activatable fluorescent probes for imaging but it’s not clear to me just how specific these probes are for NQO1 versus NQO2 and for NQO1 versus other one and two electron reductases (eg cytochrome P450 reductase, cytochrome b5 reductase, carbonyl reductases, xanthine oxidase/dehydrogenase etc).The only evidence along those lines that I could see was a reference to the use of the NQO1 inhibitor dicoumarol diminishing CL signal (line 619) but dicoumarol is itself very non-specific and this is just not sufficient evidence for NQO1-specificity. This is a central argument in this manuscript and should be expanded and clearly discussed.
In view of points 1 and 2, Figure 9b appears premature and should be deleted.
Minor points.
1) Bringing together a number of different areas has resulted in a large number of citations. Please carefully check the citation list and consider additions/deletions below. Examples;
Line 83-87 – NQO1 polymorphism 187 (Pro to Ser) needs to be adequately referenced including findings in different ethnic groups
Line 137 – Consider adding Cadenas 1995 (current ref 49) here since this work contained extensive discussion of the effect of SOD on autoxidation
Line 111 – the correct citation for NQO1 catalyzing enzymatic removal of superoxide is ref 63
2) Line 182 refers to unpublished data for apoptotic or necrotic cell death from futile redox cycling. It seems that there would be many appropriate B-lapachone or deoxynyboquinone publications (eg 51 and 53) that could be used or added. Subsequently, lines 183-189 refer to immunogenic cell death from futile redox cycling from the authors labs. Can any citations be added?
3) Line 693. The authors have covered a lot of ground in this review and have many useful points to summarize in the conclusions section. Introducing the subject of hypoxia in the conclusions seems misplaced and could be better dealt with in an earlier section.
4) Minor typos. Line 289, Delete"of" or "as". Line 303, Delete "the", Line 666-668, sentence needs to be rephrased.
Reviewer 2 Report
Comments and Suggestions for Authors
The article reviews the enzyme NAD(P)H oxidoreductase 1 (NQO1) and its role in anticancer therapies and tumor imaging. It highlights the high expression of NQO1 in cancers compared to normal tissues, making it a selective marker for neoplasms. NQO1 catalyzes the reduction of quinones to hydroquinones, consuming NADPH and generating cytotoxic reactive oxygen species (ROS) and H2O2. This quinone bioactivation strategy, due to elevated NQO1 levels in tumors, presents a unique therapeutic opportunity, though it has not yet been clinically exploited. The article further discusses the development of NQO1-targeted small molecule probes for cancer imaging, offering promise for guided cancer surgery and theranostic applications.
Comments
- There are many instances of double spaces throughout the text that need correction.
- Besides the compounds tested in their laboratory, are there other substrates of NQO1 that create a futile cycle and are related to ICD?
- The authors mention a study proposing a theranostic approach combining antitumor drug delivery and diagnostic imaging through NQO1-directed probes. This approach uses near-infrared fluorophores in human tumors (Figure A & B) and prototype compounds synthesized (reference 148). In this potential clinical application, a cancer patient would receive an intravenous injection of the NQO1-targeted fluorescence agent before surgery to guide a treatment based on photodynamic therapy (PDT). The question arises: Is photodynamic therapy capable of modulating NQO1?
- The authors should include a table of NQO1 substrates whose cytotoxicity is bioactivated by NQO1, listing their main characteristics.
